# Understanding Nutrition and Metabolism of Threatened, Data-Poor Rheophilic Fishes in Context of Riverine Stocking Success- Barbel as a Model for Major European Drainages?

**DOI:** 10.3390/biology10121245

**Published:** 2021-11-29

**Authors:** Koushik Roy, Peter Podhorec, Petr Dvorak, Jan Mraz

**Affiliations:** South Bohemian Research Center of Aquaculture and Biodiversity of Hydrocenoses, Faculty of Fisheries and Protection of Waters, University of South Bohemia in České Budějovice, Na Sádkách 1780, 370 05 České Budějovice, Czech Republic; kroy@frov.jcu.cz (K.R.); podhorec@frov.jcu.cz (P.P.); dvorakp@frov.jcu.cz (P.D.)

**Keywords:** river stock enhancement, rheophilic fish, amino and fatty acids, fish nutrition, conservation fisheries, data-deficient meta-analyses methodology

## Abstract

**Simple Summary:**

It is important to consider the nutritional requirements and optimum feeding of hatchery-raised, conservation-priority rheophilic fishes meant for river stock enhancement. Properly selected diets for these nutritionally data-poor fish species are crucial to achieve optimum growth potential and physiology. It greatly reduces the time spent in captivity, and if properly trained, may produce fish that are better insured for facing hardships in the wild.

**Abstract:**

Large-bodied, river-migrating, rheophilic fishes (cyprinids) such as barbel *Barbus barbus*, nase *Chondrostoma nasus*, asp *Leuciscus aspius*, and vimba bream *Vimba vimba* are threatened in major European drainages. This represents the subject of our present study. Their hatchery nutrition prior to river-release is mostly on a hit-and-trial or carp-based diet basis. The study demonstrates an alternative approach to decide optimum nutrition for these conservation-priority and nutritionally data-poor fishes. The study revealed barbel as a central representative species in terms of wild body composition among other native rheophilic cyprinids considered (asp, nase, vimba bream). Taking barbel as a model, the study shows that barbel or rheophilic cyprinids may have carnivorous-like metabolism and higher requirements of S-containing, aromatic, branched-chain amino acids (AAs) than carps. Besides, there are important interactions of AAs and fatty acids (FAs) biosynthesis to consider. Only proper feeding of nutritionally well-selected diets may contribute to river stocking mandates such as steepest growth trajectory (≈less time in captivity), ideal size-at-release, body fitness (≈blend-in with wild conspecifics, predator refuge), better gastrointestinal condition, maximized body reserves of functional nutrients, and retention efficiencies (≈uncompromised physiology). Considering important physiological functions and how AA–FA interactions shape them, hatchery-raised fishes on casually chosen diets may have high chances of physiological, morphological, and behavioral deficits (≈low post-stocking survivability). Based on the observations, optimum nutrient requirements of juvenile (0+ to 1+ age) barbels are suggested. Future efforts may consider barbels as a nutrition model for conservation aquaculture of threatened and data poor rheophilic cyprinids of the region.

## 1. Introduction

Freshwater fishes are the most threatened group among vertebrates, with 39% of all European fish species facing extinction [1]. Rheophilic fishes formerly dominated the fish community in the headwater sections of many European rivers (e.g., brown trout *Salmo trutta* L., European grayling *Thymallus thymallus* L., nase *Chondrostoma nasus*, barbel *Barbus barbus*, etc.). Presently, their catch is generally restricted to few individuals in such stretches [2]. The latest living planet index report for migratory freshwater fish showed a stern population decrease in Europe (−93% versus a global average of −76%), compared to what existed in the 1970s. The rheophilic potamodromous fish (migrating within rivers) have declined more (−83%) than the fish migrating between river and sea [3]. Declines have been most substantial for the gravel-spawning species of the hyporhitral and epipotamal streams in medium- and large-sized rivers (e.g., grayling Thymallus thymallus, nase *Chondrostoma nasus*, barbel *Barbus barbus*) further aggravated by their much longer generation time over smaller fishes [1]. The reasons for this decline are many, and to name a few, they are river (habitat) fragmentation [4], the effects of increasing temperature, and increasing fine sediments [5] or altered river flow and hydropeaking [6]. In central European rivers, Mueller et al. [1] identified fish species that deserve higher priority in conservation management, and half of them are large potamodromous rheophilic cyprinids. These are the subjects of the present study.

Stocking programs are being carried out to restore rheophilic fishes in European rivers with stocking materials ranging from eyed-ova stage to 1+ fish, but mainly fry [5,7,8,9]. The decision of stocking materials ultimately released to the habitats is also taken within several compromises (for example, the ability of the hatchery, the transportation apparatus, environmental conditions of the releasing area, acclimation procedures, the releasing technique, behavioral deficits or naivety of the fishes to be released, availability of sufficient natural food in the habitat, density of wild young-of-the-year fish, predator presence, days to overwintering, etc.) [10,11,12,13,14]. Besides, most hatcheries prefer to ship smaller sizes within the shortest possible time frame to keep economics and logistics favorable [10,11]. Lately, with the advent of ‘off-season breeding’, there has been some opportunity in overcoming some of these issues [14]. However, their captive feeding is mostly on second priority, hit-and-trial basis, often leading to morphological alterations and nutritional deficiencies [15,16,17,18]. They may cascade to undetected causal mortalities after release which is easy to be blamed on problems existing in the habitat or release site.

Stocking of hatchery-farm raised fish in wild rivers has not always been successful [5], and opinions are very split about the question of whether stocking can be a solution [7], especially without addressing the habitat restoration first [8,9,10]. Nevertheless, the impact on post-stocking mortalities cannot be ignored from the husbandry side. For example, too small size at stocking may impair survivability [5,11,12,13]. On the other hand, longer time spent in captivity and resulting behavioral deficits might impair survivability too [8,14]. A lack of acclimation or conditioning procedures might occur before release, (i.e., in-situ pen or cage-based rearing, pond rearing with natural food, adding complexities within hatchery rearing tanks, training to live feed, exposure to predators, and smell of injured conspecifics) [7,8,14,15,16]. Several studies confirm the ‘bigger is better’ hypothesis in the context of post-stocking chances of survivability (discussed later). There could be situations as well where it may not conform, if physical problems in the habitat have the upper hand [5,17]. A detailed discussion in this regard has been provided in the Appendix A.

The nutritional requirements of rheophilic cyprinids are not well understood. Unlike commercially important fish species with prolific nutrition research history, efforts are quite limited for rheophilic cyprinids till date [18,19]. Conventional feed selection for rheophilic cyprinids involves selection of feed that are best suited for their closest phylogenetic relative common carp (*Cyprinus carpio*) whose nutritional requirements are well established in the literature [18,20,21,22] or simply using salmonid feed that are superior to carp feed in terms of protein content [23,24,25]. Our hypothesis was that such conventionally selected feed might not be a good practice. Despite being members of family cyprinidae, rheophilic cyprinids such as barbels might have different nutritional physiology that may require better selection of diets for achieving best growth. Carp-based diets may not be the best choice at hatchery level. And in such data-deficient situations, how to select the best feed (?). The present study also follows the hypothesis ‘bigger and faster is better’ (see, Section 4 and Appendix A) in the context of riverine stocking success and considers it achievable within the shortest time spent in captivity.

Taking hints from body composition, wild food composition, and well-established nutritional requirements of a close phylogenetic relative (*Cyprinus carpio*), the present study attempted to derive a captive feeding plan for nutritionally data-deficient and conservation priority rheophilic cyprinids in central European drainages. Also, the management decisions surrounding feeding itself were optimized, taking hints from the optimum feeding conditions that exist in the wild (temperature, water flow, ontogenic diet shift) or meta-analyzing the captive growth performance under conventional rearing (diet, temperature, density, flow, etc.) from the published literature. In many aspects, the study offers novel information that could be directly applied to ongoing conservation efforts in the major European drainage systems. The proposed framework may also contribute to better hatchery nutrition decisions. Our study does not undermine the importance of habitat restoration or pre-stocking conditioning measures to be applied to riverine fish seeds but rather a synergistic tool to improve restoration successes of imperiled fish species in regional rivers using knowledge of fish nutrition.

## 2. Materials and Methods

### 2.1. Assessment and Comparison of Wild Body Composition

Juvenile to sub-adults (length range 6–25 cm) of four rheophilic cyprinids (barbel *Barbus barbus*, nase *Chondrostoma nasus*, asp *Leuciscus aspius*, and vimba bream *Vimba vimba*) were collected (minimum six individuals each) from previously known locations (either from established research stations or information from sport fisheries association members) on River Vltava and River Berounka, the Czech Republic during the middle of the vegetative season (summer; May–August). Wild fishes were electro-fished (220–250 V, 1.5–2.5 A, 63 Hz) and sacrificed with a quick blow to the head. The whole body was immediately put in iceboxes, brought back to the laboratory, made whole-body mash, and stored frozen at −80 °C. The frozen samples were sent to an accredited third-party laboratory (Agrola s.r.o.) for the analyses of dry matter, crude protein, crude lipid, crude ash, crude fiber, altogether called proximate composition; according to ISO certified methods (e.g., ISO 11465:1993, ISO 16634-1, ISO 1443:1973, ISO 1575:1987, ISO 5498:1981). Additionally, essential amino acids (EAA), non-essential amino acids (NEAA), and total phosphorus (P), were analyzed (e.g., ISO 6491:1998, ISO 13903:2005). Amino acids included methionine, lysine, threonine, aspartic acid, serine, glutamic acid, glycine, alanine, tyrosine, valine, phenylalanine, isoleucine, leucine, histidine, arginine, cysteine, proline, tryptophan. They were analyzed following HCl digestion. ninhydrin derivatization and ion chromatography. Fatty acids analyses were carried out in our own laboratory following hexane-isopropanol extraction, fatty acid methyl esterification, and gas chromatography with flame ionization detection [26].

Biochemical parameters of individuals were pooled species-wise. Any significant inter-specific difference (α level set at 0.05) in body composition was assessed by statistical test. Each parameter was first subjected to a Shapiro–Wilk’s normality test (SWN test). Then, following the *p*-value of SWN test (normally distributed data if *p* > 0.05; not normally distributed data if *p* < 0.05), either parametric test for normally distributed data (one-way ANOVA with post-hoc Tukey HSD) or non-parametric test for not normally distributed data (Kruskal-Wallis post-hoc Dunn’s test with Bonferroni correction) was selected. Most biochemical parameters were normally distributed. Only few parameters were not normally distributed owing to skewness of data; majority of them were from energy, lipid, and fatty acids category. Among native rheophilic cyprinids considered, species with statistically similar (*p* > 0.05) body composition were assumed and classified to be similar in terms of nutritional requirements. The tests were performed using RStudio v1.2.5042 using libraries ‘e1071’, ‘dunn.test’ and ‘FSA’ [27,28].

### 2.2. Selection of Model Species and Calculation of Target Nutrition

The authors selected barbel as a model species to study due to its similarities in body composition with other native rheophilic cyprinids considered (asp, nase, and vimba bream) in Central European rivers [1,2,29]. Compared to asp, nase, and vimba bream, the barbel also had relatively better data availability in terms of wild growth, food, and feeding habits or captive feeding experiments.

Three principally different approaches were used to calculate the target nutrient range, hereinafter referred to as the ‘target range’. The first (‘Approach A’), mimicking the nutrient composition of natural food for the given species, is meta-analyzed from several studies (see Appendix A). Calculations for ‘Approach A’ are further detailed in the Appendix A. For the second (‘Approach B’), dietary nutrition levels are back-calculated from the body composition of given species and fitted with a typical cyprinid nutrient retention scheme. Calculations for ‘Approach B’ are further detailed in the Appendix A. Lastly (‘Approach C’), standardized nutrient specifications [18] of a phylogenetic relative (*Cyprinus carpio*, common carp) are used as status quo choices. Further details about ‘Approach C’ can be found in the Appendix A. The precaution with either of these approaches is also briefed in the Appendix A.

### 2.3. Selection of Experimental Diets and Feed Preparation

Based on our calculations on barbel (explained above), a target nutrient range was calculated (Figure 1). The nutrients of priority, for which the target range was calculated, were protein, essential amino acids (EAA; n = 10), non-essential amino acids (NEAA; n = 8), lipid, and phosphorus [30,31,32]. All selected diets easily fulfilled the required levels of omega-3 (n-3) and omega-6 (n-6) fatty acids (FAs) (usually 0.5–1.0% of the diet) [18]. Hence, the evaluation of results focused less on fatty acids. Around this ‘target range’, premium diets from commercial aquafeed manufacturers were screened and selected. Four commercial diets were selected (diet A = Aller Futura EX GR 0.5–1.0 mm; diet B = Skretting ME-1.0 MP Presta; diet C = Skretting ME-3 Meerval Top; diet D = Skretting C 4 Carpe-F) for feeding through mid-0+ age and early 1+ age. Diet A had protein 51.66%, lipid 12.41%, ash 10.46%, fiber 8.79%, nitrogen-free extract (NFE) 16.68% and gross energy 3850.5 kcal kg^−1^. Diet B had protein 50.87%, lipid 12.65%, ash 7.83%, fiber 8.96%, NFE 19.69% and gross energy 3960.9 kcal kg^−1^. Diet C had protein 45.93%, lipid 7.10%, ash 14.30%, fiber 10.50%, NFE 22.20% and gross energy 3364.2 kcal kg^−1^. Diet D had protein 36.70%, lipid 7.06%, ash 7.96%, fiber 6.44%, NFE 41.85% and gross energy 3777.4 kcal kg^−1^. Further detailed composition up can be found in Figure 2. Declared ingredients in the commercial diets (without a recipe; proprietary information) are listed in Appendix A.

The diets were selected in a way that two diets (diets A, B) ‘fulfill’ the target range hereinafter called optimum diets, while one diet (diet C) fell ‘just below’ or ‘near’ the target range called the average diet. Lastly, one diet (diet D) was ‘well below’ target range called the sub-standard diet. Feed A and B were better choices than conventional carp-based diets. Feed C and D were conventional carp-based diets usually selected in the hatcheries, with feed D being most low-cost option. Since the commercial pellets were of varying size and having different surface textures, they were re-pelletized to have similar starting conditions (pellet size, texture) to avoid any unforeseen interferences caused by them; this is detailed in the Appendix A.

### 2.4. Growth Trial-I (Mid 0+ to Late 0+; 100 Days)

Hatchery-raised 0+ barbel (=5 months post-hatching and onset of exogenous feeding) were stocked in a series of laboratory maintained, well-screened, translucent lid, flow-through RAS systems (water volume 60 L tank^−1^; flow 3 L min^−1^ or 300% of tank volume per hour), with established daily cleaning (flushing), temperature (limits = 21.5–23.7 °C), dissolved oxygen (≥75% saturation), pH (6.6–7.9 units) maintenance protocols and periodic unionized ammonia checks in the water (<0.05 mg L^−1^). Four experimental groups (Diets A, B, C, and D), with triplicate per group, were allocated (4 × 3 = 12 tanks). Fishes were starved for 48 h to get rid of any residual food in the gut. Per tank, barbels were stocked at an initial stocking density of 3.83 ± 0.2 kg m^−3^ (i.e., 230 ± 12 g per 60 L or tank). The stocked fish had a total body length of 7.7 ± 0.5 cm (interquartile range IR: 7.4–8 cm; coefficient of variance CV: 7%) and body weight 3.9 ± 0.8 g (IR: 3.4–4.4, CV: 21%) and reared for 100 days. Fishes were fed manually with experimental diets (A, B, C and D) at 6% of standing tank biomass per day. The daily feed ration was divided into three equal and smaller split doses (i.e., 2% each at morning (~07:00 HRS), noon (~11:00 HRS) and afternoon (~15:00 HRS)). Following intermittent measurements, the daily feed ration was revised with increasing tank biomass to keep feeding dose (6% of tank biomass) consistent. After each feed dosage revision, the amount of uneaten feed flushed out of the tanks were qualitatively monitored for five days by expert technical personnel. Since the feed was given in three smaller split doses, each time the uneaten feed could be monitored easily (against feces particles) and logically summed up for the whole day. However, an accurate measurement on dried biomass was not conducted due to suspended fines losses, dissipated particles of feed pellets, occasional mixing with fecal matter in the solid accumulation chamber. On an average, at the beginning of experiment, 5% of uneaten feed after each feed application amounting to ~15% of uneaten feed losses per day was apparent. It corresponds to ≈5.1% of biomass eaten out of 6% of biomass feed applied. As the fish (biomass) grew and applied feed was accordingly revised to keep consistent feeding rate through the experiment, uneaten feed losses increased to ~10–12% per feed application time (on an average); cumulative to ~30–36% uneaten feed losses by the end phase of experiment (≈3.8–4.2% out of 6% of biomass eaten). Therefore, the average of apparently eaten feed through the experiment lifetime (rounded-off; ~4.5% of biomass) was used for calculations on nutrient utilization.

Feeding was suspended for 24 h before any biometric measurement. At the start and end of the trial, six starved fish (devoid of feed pellets in the gut) from each group were sacrificed and immediately frozen-stored (−80 °C) to be later used for whole-body proximate composition analyses in an accredited third-party laboratory (mentioned as above). On completion of growth trial-I, fishes were 8+ months old. Statistically significant differences in growth and nutrient utilization parameters (for either growth trials) among different fed groups were assessed by parametric or non-parametric test following a normality test. The basic procedure is described above (section on body composition).

### 2.5. Validation Growth Trial-II (Late 0+ to Early 1+; 50 Days + 36 Days + 64 Days)

Fishes (age 8+ months) from the worst-performing diet group (diet D followed by C) were purposively selected for the second round of the growth trial. Proximate composition of the diets can be found in the previous section. The main growth trial (64 days) was preceded by two arbitrary trials (50 days + 36 days). The purposes were: (a) to observe any compensatory growth mechanism if these fishes are switched to a proven optimum diet from the previous round (50 days arbitrary trial; daily feeding ~3% of body weight; final length-weight measurement); (b) to quantify growth when an optimum feed is fed at a basal/maintenance feeding rate (~1.5% body weight per day; 36 days arbitrary trial; final length-weight measurement); and (c) a full-fledged growth trial for 64 days, validating the trends in growth performance under diets A, B, C, and D, observed in the previous 100-days growth trial. The same system and conditions (outlined above) were maintained for the growth trial.

Barbels were stocked at a 7.7 kg m^−3^ per tank (i.e., 460.4 g per 60 L or tank). Stocked fish had a total length of 14.5 ± 2.2 cm (IR: 12.8–16 cm; CV: 15.2%) and body weight of 29.1 ± 13.8 g (IR: 18–37.6 g; CV: 47%). Daily feeding rate was 4% of biomass, applied in three equal and smaller split doses. The biometric measurements (post-24-h starving) were done exhaustively on all fishes (per tank) and total tank biomass was also taken. As in growth trial-I, intermittent growth measurements were done through the course of the experiment and daily feed ration was revised accordingly keeping the feeding rate consistent (at 4% of body weight). Following each feed dosage revision, the amount of uneaten feed flushed out of the tanks were qualitatively monitored for five days by expert technical personnel (detailed under trial-I). At the beginning of the experiment, the uneaten feed was quite low; on an average ~2% of uneaten feed per application time, amounting to ~6% uneaten feed per day (≈3.7% out of 4% biomass eaten). As the fish grew, the uneaten feed per application time increased to 4% (on average) or 12% uneaten feed per day (≈3.5% out of 4% eaten) towards the end of the experiment. Considering the average of apparently eaten feed (3.6% of biomass) through the course of the experiment, calculations on nutrient utilization were done. Fishes were sacrificed at the beginning (six baseline fish) and end of the trial (three fishes per tank), after 24 h starvation, for whole-body proximate composition (mentioned above). Six larger sized individuals were also dissected to assess physiological well-being after different commercial feed treatments. On completion of validation growth trial-II, fishes were 13+ months old.

### 2.6. Evaluation of Growth, Factors Affecting Growth, and Physiological Performance

#### 2.6.1. Evaluation of Experimental Results

Parameters such as growth trajectory (thermal growth coefficient, length increment per day, length-at-age), body fitness (Fulton’s condition factor), feed utilization efficiency (feed conversion ratio, protein efficiency ratio, retentions, losses), and nutritional physiology (body nutrient composition, nutrient retention ratios, gastrointestinal, and liver morphology) were assessed. Other observations such as density- and size-dependent growth, compensatory growth mechanism, yield, economics, and size suitability for riverine stocking were also made. Details on the parameters can be found in the Appendix A.

#### 2.6.2. Retrospective Evaluation against Reviewed Metadata

A systematic literature search for peer-reviewed scientific articles on barbel *Barbus barbus* was carried out through online databases (google scholar, scopus and web of knowledge). Two separate searches were conducted. Once for wild data and once for captive data. The collection of information followed the scheme of identification → screening → eligibility → inclusion [33]. Altogether, 52 articles were included in an exploratory meta-analysis. The procedure is summarized in more detail in the Appendix A.

Published data on wild or captive growth including parameters such as length-at-age, length increment, Fulton condition factor, feeding conditions (temperature, flow, body size, feeding ration, nutrition levels, density, ontogenic diet shift) were compared with the present observations. More specific information on retrospective evaluation of obtained results can be found in the Appendix A, including a synthetic index called ‘riverine stocking suitability’.

## 3. Results

### 3.1. Assessment and Comparison of Wild Body Composition

Detailed body compositions of four native rheophilic cyprinids commonly occurring in Central European drainage systems are provided in Figure 1. Based on the hetero-specific comparison, barbel may be a central representative species with statistically similar (*p* > 0.05) body nutrients composition with all or most of the rheophilic cyprinids considered (asp, nase, and vimba) (Figure 1). Protein, almost all amino acids, 11 out of 21 fatty acids, phosphorus, and non-protein energy balances (with protein or gross energy) were comparable (Figure 1). Within conspecifics, parameters such as dry matter, protein, and most amino acids (AAs) seem stable, with an average coefficient of variation (CV) less than 6%. Parameters such as ash, phosphorus, P: N ratio, gross energy, arginine, tyrosine, and some long-chain fatty acids (within C20 to C22; Figure 1B) had CV between 7–11%. Lipid, some shorter chain fatty acids (within C14 to C18; Figure 1B), carbohydrates, and energy had average CV > 12%.

Pearson’s 2-tailed correlation between mean body AAs and fatty acids (FA) (Figure 3) revealed: (a) significant negative correlations (*p* < 0.05) between branched-chain AAs (isoleucine, valine) and long-chain FAs (C18:0, C20:0, C20:2n-6, C20:3n-3), (b) significant positive correlations (*p* < 0.05) between arginine and some fatty acids (C16:0, C16:1, C20:5n-3), and (c) significant positive as well as negative correlations (*p* < 0.05) between cysteine and some unsaturated fatty acids (C18:1n-7, C18:2n-6, C22:5n-3). The amino acids and fatty acids contents in body can have seasonal variations. Present results may be considered valid for individuals during summer months (May to August), i.e., peak vegetative season in the temperate, Central European Rivers (located at elevation 310–318 m asl). Proposed ideal protein concept and optimum lipid balance derived from barbel body amino acids and fatty acids profile, respectively, may be deemed valid for actively growing fish.

### 3.2. Optimum Nutrition for Achieving Steepest Growth Trajectory

Barbel’s target nutrient range was calculated using three principally different approaches (Figure 2). Comparing the approaches used, neither one of them is safe enough individually. All three approaches, when considered altogether, may be a safer and valuable tool for calculating the target range of nutrients in the diet. For example, while approach ‘A’ has not underestimated protein levels, there is a potential under-estimation of essential amino acids (EAAs) methionine and lysine. Whereas approach ‘C’ potentially takes care of EAAs methionine and lysine but under-estimates protein and other EAAs histidine and leucine (which was projected adequately by ‘A’) (Figure 2). In a nutshell, the approaches complement each other. The median of the target range or the target range itself can be used as the baseline for diet selection.

Selected diets which fulfilled our target range (diets A and B) performed best. They had superior growth (Table 1 and Table 2), length increment (Figure 4), body fitness (Figure 5), and extraordinary size-at-age or growth trajectory (Figure 6). The final body size or size-at-age (≈20 cm total body length in ≤1+ age) or body fitness (equivalent to top 25% fit individuals in the wild) was such that the cohorts were almost or exactly suitable for riverine stocking, in the sense that they may easily blend-in with bigger and fitter conspecifics in the wild (Figure 7). The body composition (Figure 8) was also affected by the nutrition regime. Optimum diets could produce fishes with higher body energy reserves and good body protein levels. When such fish is subjected to starvation during pre-stocking acclimatization or conditioning or hardships in the wild, they might be better insured.

Diets falling below calculated target nutrient range i.e., orange to red cells in Figure 2, performed moderately (diet C) to poorly (diet D). Some visual effects of poor feeding choices in the early life stages can be seen in plates showing hepatopancreatic lipid reserves (Appendix A) and physique (Appendix A). Analyses of the liver revealed that from sub-standard diets (diet D) to optimum diets (diets A, B), saturated fatty acid (SFA) reserves increased, while polyunsaturated fatty acid (PUFA) reserves slightly decreased in liver but greatly deposited in muscle (Figure 9). Particularly, the muscle reserves of some essential fatty acids such as omega-3 fatty acids, EPA (eicosapentaenoic acid), and DHA (docosahexaenoic acid) increased from sub-standard to optimum diets (Figure 9). In fact, the bigger individuals under diets A and B developed immature but conspicuous gonads. Diets A and B also had the highest DHA:ARA ratio (docosahexaenoic acid to arachidonic acid ratio > 16:1) and DHA: EPA ratio (docosahexaenoic acid to eicosapentaenoic acid ratio above ~1:1) compared to diets C and D (DHA:ARA < 6:1; DHA:EPA < 0.7) (Figure 2B). The effects on gastro-intestinal lumen (microvilli) can be found in the Appendix A (Appendix A). Poor diets probably manifested onto lower gut enzymatic activity (indirectly assessed), besides thinner and longer intestine (qualitative observations).

In a nutshell, results from the growth trials validate our approach of target nutrient range calculation and categorization or selection of diets around the calculated target nutrient range. Within the limitations of our selected diets, our results hint at an indicative crude protein, amino acids, phosphorus, and fatty acids (optionally) range that may be suitable for raising barbel juveniles in the context of riverine stocking. An ideal commercial diet can be selected mimicking nutrition levels in diet A (ideally) or at least diet B (i.e., following green cells in Figure 2) for hatchery raising of 0+ or 1+ barbel juveniles. It would help to achieve best possible size-at-release within shortest possible time. For example, approximately ≈ 20 cm total length by ≤1+ age was achieved, that too with good Fulton’s condition factor (body fitness). It is usually achieved in 2+ or 3+ year age groups under conventional hatchery raising or rivers, respectively (Figure 6 and Appendix A). For achieving a steep growth trajectory or realizing full growth potential of these species (Figure 6 and Figure 7), both optimum feed and feeding conditions are crucial (presented below).

### 3.3. Understanding Nutritional Physiology and Conditions for Achieving Maximum Growth

Within our graded levels of diet selection (Figure 2 and Appendix A), protein, lipid, phosphorus are the closest and significant drivers of thermal growth coefficient (TGC; hereinafter referred to as ‘growth’) (Figure 10). As expected, the growth increased with increasing dietary amino acids and fatty acids (Appendix A). Comparing the proximity of the TGC vector with amino or fatty acids vectors (Appendix A), TGC may have responded most strongly (compared with other AAs) with increasing proportions of sulfur-containing AAs (methionine + cysteine) and aromatic AAs (phenylalanine + tyrosine) from diet D to Diet A. Also, diets with higher omega-6 fatty acids (n-6 FAs) did not seem to have negative repercussions on growth; a positive correlation was found (Figure 10). Dietary non-protein energy to protein ratio (NPE:P) ratio, and carbohydrate content were negatively correlated with growth; their contribution was strong (Figure 10). Protein (N) and P retentions were positively correlated with growth (Figure 10, Table 1 and Table 2). They were also significantly higher (*p* < 0.05) under diet A which comfortably fulfilled or fell well above the target nutrient range (Table 1 and Table 2). When dietary amino acids are well above the target nutrient range, protein efficiency ratio by fishes are significantly (*p* < 0.05) improved (e.g., diet A; Table 1 and Table 2). The NPE:GE retention ratio was negatively related to growth (Figure 10).

Despite providing diets with near optimum nutrition levels, proper pre-conditions for efficient feeding are often not taken care of. Or proper feeding conditions are provided but without optimum diets. If poor diets are mistakenly selected in the hatcheries initially, the mid-0+ to early 1+ fish still offer something called ‘compensatory growth mechanism’ when switched to an optimum diet (Figure 11). However, the nutritional distance between the diets must be significant (diet B significantly better than diet D, *p* < 0.05; Figure 11). Achieving optimum growth also depends on feed rationing; optimum feeding maybe at 4–6% of body weight (Appendix A). Growth trajectory usually dampens around 12–18 cm total body length or 17–19 kg m^−3^ density. However, an optimum diet may help to delay such dampening, and optimum growth rate is realized longer. Besides, selection of summer-like water temperatures through the rearing period (~22–23 °C) and mild flow conditions to enable feeding (~3 L min^−1^ or 300% water exchange per hour) are necessary.

## 4. Discussions

### 4.1. Bigger and Faster Is Better for Riverine Stocking Success

A detailed discussion on the scientific background of our ‘bigger and faster is better’ hypothesis in the context of riverine stocking (e.g., a refuge from size-selective predators; swimming endurance) is provided in the Appendix A. Our hypothesis is partially modified to Sogard et al. [34] since hatchery-raised fish have undesired domestication effects under prolonged captivity. For example, behavioral naivety [8,14]. Some common solutions to improve post-stocking smartness of hatchery raised fish is discussed in the Appendix A.

### 4.2. Status Quo Feeding Management of Riverine Fish Seeds and Their Fallacies

Unfortunately, many threatened, conservation-priority fish species are nutritionally data deficient; their nutritional specifications do not exist [18]. Large-bodied rheophilic cyprinids native to the Central European drainage systems (for example, barbel, nase, asp, vimba bream are such examples) [1]. For these nutritionally data-deficient fish species, the feeding decisions in hatcheries are often on a hit-and-trial basis or assumption basis. For example, >51% protein diet at 17 °C in Kaminski et al. [35] or 33% protein diet at 19.5 °C in Policar et al. [22] given to 0+ or early 1+ barbels, resulting in far less growth than what can be achieved through smart feeding of optimum diets at 4–6% of body weight (Figure 4). An arbitrary combination of dry feed and/or frozen chironomid larvae have been recommended for 0+ barbels [20,35,36]; probably taking a hint of barbel’s natural food in the wild [37]. Such natural prey (on wet weight basis) could barely result in TGC of 0.26 units in 0+ barbel (re-calculated from [35]). Same TGC is achievable when fed a sub-standard diet (on dry weight basis) to near satiation (4% of body weight) (diet D; present study). Over the past, feeding of barbel have included commercial salmonid, catfish starter diets having adequate crude nutrient levels (protein > 50%; lipid < 15%; phosphorus < 1%), but also at a restricted feeding rate (1–3% body weight) or sub-optimum temperatures (<20 °C) [22,23,35,38,39,40]. Such inconsistencies in diet selection may be partly due to unavailability of commercial feed that are tailored or recommended for these species. It keeps the hatchery managers open to resort to multiple or best available option in the market. Besides, low temperatures (≤20 °C; [22,35,39], too slow flow (0.2–0.3 L min^−1^; [35,38,40] or rapid flow (6–10 L min^−1^; [22,23] did not harness the maximum growth potential in this species (Figure 4). Smart feeding of optimum diets should also include proper temperature (~22–23 °C) and flow (300% water exchange per hour). Compared to wild conditions, the growth under conventional feeding in captivity is just comparable to slightly better (Figure 4, Figure 5 and Figure 6).

Growth in the wild occurs within bottlenecks of thermal habitat, feed availability, and sometimes metabolically stressful water flow. For example, most typical European rivers, with barbel zones, have a thermal regime between 4–25 °C [41,42]. Barbel zones have flows > 10 cm s^−1^, much higher than other cyprinids prefer [43,44]. Temperature regime for active growth and feeding in European rivers range between ~15–19 °C, that lasts for only ~4–5 months a year [41,45,46,47,48,49,50]. Moreover, barbels have a lower threshold temperature for growth (13.5 °C), below which they stop growing [41]. Temperate common carp strains, a phylogenetic relative of barbel, are known to lose appetite even before such temperatures are reached [51]. Sometimes, barbels lack protein and organic phosphorus-rich primary food items such as benthic macroinvertebrates and must resort to feeding plant matter [52], which has low protein, high fiber, and largely indigestible phytate P [51]. Fishes are also reported to elongate their gut length in response to such a low-protein and plant-dominated diet over extended periods for increasing absorptive surface area, digestive efficiency, and food residence time in the gut [53,54]. Such processes are metabolically costly [54], and they might also occur with a sub-standard diet (e.g., diet D). Wild barbels would sometimes also spend much more energy maintaining their position in the fast-flowing streams than they derived from food [55]. Hence, achieving a steep growth trajectory (i.e., extraordinary size-at-age within the shortest possible time in captivity) might not be possible under the status quo approach. Even if the growth of juveniles (0+, 1+ fish) are comparable to slightly better than the wild, it may not be enough in the context of riverine stocking.

Rearing of river stocking materials to larger sizes often has unintended adverse effects. When unscientifically chosen feed is used, there is a high chance of nutritional deficiencies or morphological alterations [56,57,58,59]. Although morphological alterations are visible to the naked eye, physiological alterations or deficiencies in the body’s nutrients pool remain subtle. Post-stocking survivability is below 10% for both smaller size-at-release (fry; [5]) or bigger-size-at-release (advanced fingerlings; [15]). The conditioning processes prior to release or good habitat conditions at release site are decisive factors for post-stocking survivability. However, the effects of poorly, unscientifically chosen captive feeding cannot be ruled out; the results of the present study point in this direction. For example, barbels raised on average to sub-standard diets had a gain in body length without a proportionate gain in weight (≈poor fitness). It also happens in the wild and is probably indicative of poor growth conditions [60]. After release, such individuals would most likely perish due to competition from extant barbel and carp populations [61]. When barbels starve, they use their lipid reserve for energy while preserving protein, ash, and water to maintain body mass (Figure 8). It is an adaptation for submerged-buoyant animals, such as fish, not to disturb their center of gravity and thus orientation [62]. There is a pattern too in this; lipids classes are preferentially catabolized in the following order, from the body reserves: SFA → MUFA → PUFA [63,64]. Barbels raised on average to sub-standard diets had lower essential lipid reserves. Lipid reserves in the body act as ‘insurance’ during challenging situations in the wild (e.g., surviving high flow, food deficit, migration, and spawning decisions) [43,52,55,65,66]. Additionally, dark reddish or brownish liver color (presently observed) is associated with low lipid reserves in fish liver [67], which might be ideal from an aquaculture perspective, but maybe counter-productive from a riverine stocking perspective. If stocked in the rivers, individuals raised on average to a sub-standard diet might be less insured in challenging situations and most likely perish. It opens up areas of future validation.

### 4.3. Importance of Optimum Nutrition and Metabolism for River Stocking Cohorts

In fish nutrition, optimum nutritional requirements are usually assessed through systematic dose-response experiments of individual nutrients, keeping other nutrients in the diet fixed. Nutritional requirements are then standardized for a given fish species, using meta-analyses tools on several accumulated studies (e.g., broken-line regressions for optimum requirement). It takes years of continuous research [18]. For these declining and commercially unimportant fish species, both time and interest are constraints, respectively. In the absence of standardized nutrient specifications for this data-deficient (nutritionally) but conservation-priority fish species (like barbels), the current approach and its conclusions are practical alternatives or indicative solutions. We propose that the plan for barbel may be representative for other native, nutritionally data-deficient rheophilic cyprinids such as asp, nase, or vimba bream having comparable body composition. It is subject to further validation and future research. Using a systematic framework, present study showed how to approximately design optimum captive feeding for barbels. The present recommendations do not replace the need to standardize the nutritional specifications for these less-explored fish species. The present findings narrow down the target nutrient range around which standard dose-response experiments may be conducted in the future. Future research may focus on validating barbel’s identified nutrient requirements and optimum feeding conditions for hatchery raising of other rheophilic cyprinids native to the Central European drainage systems.

Based on observations, juvenile barbels seem to be inclined towards a carnivorous fish or salmonid-like metabolism [19,68,69], despite being a member of the cyprinid family. Under low protein (or amino acids) supply, the lipids and carbohydrates are not utilized efficiently; instead, conserved in the body as non-protein energy (NPE). Protein (amino acids) and phosphorus requirements were apparently high, while the non-protein energy is preferred from lipids rather than carbohydrates. Requirements of phosphorus (P), in particular, need additional focus while diet selection [30]. Unlike calcium (Ca), supply of P is majorly reliant on food [70,71]. Hence, diets in hatcheries must be chosen wisely. The obvious ‘carp compatible diets’, due to comparative phylogenetics [22], must be reconsidered for rheophilic cyprinids of Central Europe.

Lysine and/or methionine form the central AAs around which the ideal protein concept for fish revolves since they are the first-limiting AAs in most artificial diets for fish [72,73]. The present study hinted similar importance of S-containing AAs in barbels (methionine + cysteine) and additionally aromatic AAs (phenylalanine + tyrosine) for growth (Appendix A). S-containing AAs perform various metabolic functions [31,74]. Recent evidence suggests their connecting link with fatty acids metabolism via stearoyl-CoA desaturase-1 (SCD) enzyme [75,76]. SCD is a lesser-explored enzyme than fatty acid desaturase enzyme (FAD). SCD is an emerging focus for nutritional programming in fishes; to drive lipid metabolism towards PUFA sparing and increasing their reserves in fish body [77]. In rheophilic fishes, our correlation report gives preliminary hint that S-containing AAs may indeed have a role in fatty acid metabolism, especially in de-novo bioconversions. For example, opposite correlations of cysteine with FAs 18:2n-6 (linoleic acid, LA) and 22:5n-3 (docosapentaenoic acid, DPA) are interesting. Since the activity of SCD is thought to be positively related to cysteine and inversely related to PUFAs [75]. Circumstantial evidence suggests n-3 DPA is an intermediary product between EPA and docosahexaenoic acid (DHA) [78]. Biosynthesis of EPA and DHA in fish involves sequential desaturation and elongation of FAs 18:2n-6 (LA) and 18:3n-3. Particularly, freshwater fish (such as barbels) have much higher evolutionary pressure to endogenously produce EPA, DHA in contrast to marine fish species [79,80]. In the present experiment, barbels with highest reserves of muscle EPA and DHA were fed on diets having highest levels of S-containing AAs such as cysteine (e.g., diet A, B); with a probable up-regulation of SCD activity. Therefore, for rheophilic cyprinids, understanding the interactions of dietary S-containing AAs with fatty acids synthesis is quite important. Especially when they are to be released to the wild without compromising their competencies for fatty acids or protein synthesis.

Aromatic AAs (phenylalanine + tyrosine), in addition to their conventional roles as key precursors of hormones and neurotransmitters [31], have recently been recognized for stress abatement in fish, particularly phenylalanine [81]. Due to their dynamic and fast-flowing habitat, rheophilic cyprinids may depend more on such stress abatement AAs. Tyrosine is known to control pigmentation in fish [82,83] and is also a molecule having strong antioxidant capacity [84,85]. The flowing nature of rheophilic fish habitats such as hyporhitral or epipotamal zones [1] also bring higher dissolved oxygen levels (DO). For example, Danube river DO can range from 6 to 16 mg L^−1^ within a temperature range of −3 to 30 °C [86,87]. Besides, rheophilic fishes have higher metabolic energy requirements, swimming activity, endurance, and aerobic respiration demands [17]. In such a situation, the chances of respiration-induced reactive oxygen species (ROS) formation and oxidative stress may be higher [88]. If diets do not enable sufficient reserves of these AAs in the body, succumbing to oxidative stress or compromised pigmentation (less effective camouflaging) may occur. The study has proposed optimum dietary crude AA levels (i.e., green cells in Figure 2A) and ideal protein concept [73] for juvenile barbels. Most importantly, considering all the 18 AAs together without discriminating between essential or non-essential ones (proposed paradigm shift; [32,89,90], is important to achieve extraordinary growth trajectory and significantly reduce the time required in captivity).

AS with protein sparing, the concept of omega-3 FAs sparing has emerged too [64,91], with saturated fatty acids (SFA) relative to monounsaturated fatty acids (MUFA) being of central importance. Our present observations might be hinting in this direction too. Higher SFA reserves in livers of barbels fed optimum diet sufficiently spared omega-3 FAs and allowed plenty of deposition in higher amount in muscles (including EPA + DHA) (Figure 9). It was probably conducive given that some bigger individuals (~19–20 cm) at 13+ months age developed conspicuous gonads. Such early sexual maturation in captive 1+ barbels at >20 °C and fed optimum diet (>50% protein, >12% lipid) was previously reported [25]. Nonetheless, under pre-stocking conditioning or post-stocking starvation, fishes from sub-standard diets having low SFA reserves (in the liver) and marginal polyunsaturated fatty acids reserves (PUFAs; in the flesh) would be forced to catabolize the essential, functional lipids for meeting metabolic energy demands [63]. As such, their essential reserves might be quickly lost compared to fishes fed with optimum diets and having large reserves. The present study also proposed an optimum lipid balance for barbels with an SFA called palmitic acid occupying the central position; closely following the idea of ‘ideal protein concept’ with lysine occupying the central position. Palmitic acid (C16:0) was selected as central FA for being the most common and reliable FA among the spectrum (see, Figure 1 part B), and the backbone of most widespread lipid modification called protein palmitoylation [92]. Particularly, palmitic acid (PA) to PUFA ratio is known to be a decisive factor in the occurrence and intensity of DNL [93]. The same is probably true in barbels. The trend of PA: PUFA ratio of diets (diet C 0.69:1 > diet A 0.68:1 > diet B = 0.67:1) superimposed with the trend of intensity of DNL occurrence among the fed groups (diet C > diet A > diet B). Fishes to be stocked in rivers may need to have some excess lipid or fatty acids reserves (contrary to pure aquaculture) before undergoing harsh pre-stocking conditioning or post-stocking food deprivation. In such context, these physiological loopholes (e.g., PA:PUFA ratio, specific AAs) may be targeted for valorization (e.g., promotion of DNL, higher biosynthesis of omega-3 FAs, EPA, and DHA).

Another aspect is protein-lipid-energy interactions. Branched-chain AAs (BCAAs; such as isoleucine, valine), besides their recently asserted role in glucose metabolism [94,95], are also linked with fatty acids metabolism. Recent evidence suggests catabolism of these BCAAs contributes to odd chain and long-chain FAs synthesis [96,97,98]. The presently observed negative correlation between BCAAs and some FAs in rheophilic cyprinids body (barbel, nase, asp and vimba bream; Figure 3) is interesting. It might suggest the existence of these mechanisms in high energy demanding rheophilic fishes too, eating protein-dominated and carbohydrate-limited diet in the wild. Recently, arginine has also been linked to upregulate fatty acids synthesis in fish [99,100]; positive correlation in body was also observed (Figure 3). In diets A and B, the supply of these abovementioned BCAAs and arginine was higher which coincided with higher body energy, lipid reserves (Figure 8), as well as higher muscle reserves of omega-3 FAs, EPA, and DHA (Figure 9). It hints that abovementioned interactions and physiological processes might be active in barbels too. The arginine-fatty acids nexus, in particular, is capitalized in human clinical nutrition for immunity-enhancing effects [101]. Therefore, for shaping the energy metabolism, fatty acids reserves, and immunity of juveniles meant for riverine stocking, a sufficient and well-balanced protein (amino acids) is quite important to consider.

### 4.4. Management around Optimum Nutrition to Achieve Goals of Riverine Stocking

If fishes are to be stocked in rivers, average to sub-standard feeding should be avoided at all costs. If such choices are mistakenly made, juveniles may offer a compensatory growth mechanism to compensate the nutrients or growth that was missed (Figure 11). The responsible mechanisms behind this are usually hyperphagia and advanced feed utilization efficiency [38,102]. However, the residence time in captivity may get prolonged in such situation (or maybe not if the onset of compensatory growth mechanisms is properly timed) [103]. Barbels in European rivers are known to undergo an ontogenic shift in feeding intensity (≈growth rate) and feed preference (≈lower nutritional requirement). Around 16–20 cm body length, feeding intensity is reduced to mild levels while herbivory increases [52]. It reinforces our observations on dampening of growth rate around such size ranges (identified; see results). The management should also avoid continued feeding with optimum diets if some threshold body sizes or densities (identified; see results) dampening growth are already reached. Rather, decisions on size grading, stock thinning or release to rivers could be taken. European hatcheries had raised 1+ barbels at densities up to 25 kg m^−3^ [104]. Densities of 0+/1+ barbels in Policar et al. [22] were even ≈30 kg m^−3^, and resultant TGC was only between 0.2–0.4 unit (re-calculated). Present study identified reasonable densities below which higher TGC can be expected under optimum captive feeding.

### 4.5. Future Directions

We understand that optimum captive feeding alone may not be the key to improving stocking success in rivers. Habitat restoration is simultaneously necessary. Moreover, the domestication effects or behavioral naivety should be overcome by acclimatizing hatchery fish to harsh wild conditions before stocking. While habitat restoration is a topic in its entirety, some pre-release conditioning measures are exemplified in the Appendix A. Conditioning measures should be mandatorily applied on fishes raised under optimum captive feeding which are meant for riverine stocking. It would help to produce cohorts that are morphologically, physiologically as well as behaviorally superior to average cohorts raised on conventional captive feeding or not properly trained prior to release. As a responsible code of conduct, the superior hatchery raised cohorts should be released in localities where they do not outcompete the wild, smaller young-of-the-year conspecifics [7]. Future research may focus more on these directions.

In terms of nutrition and physiology, the present study has hinted that barbel may be considered as a representative species among other native rheophilic cyprinids in Central European drainage systems. Additionally, we found that data on natural food, feeding habits, wild growth patterns, and attempts on captive feeding are much more available on barbel compared to nase, asp, or vimba bream. Based on the observations, dietary nutrient levels ‘above’ diet B or ‘ideally like’ diet A (green cells in Figure 2A,B) could be optimum nutrient requirements of juvenile (0+ to 1+ age) barbels. Capitalizing on the present findings and the information already meta-analyzed, future efforts may try to validate ‘barbels as a nutrition model’ for conservation aquaculture of imperiled and data poor rheophilic cyprinids of the region. The present research can provide a basis for further research in the area.

## 5. Conclusions

Rheophilic cyprinids in Central European drainage systems are declining fast and riverine stocking with hatchery raised fish is often carried out. There is a need to revisit conventional feed selection strategies in hatcheries raising rheophilic cyprinids for artificial stock enhancement in imperiled rivers. Better diets that are tailored for these target fish species are very crucial if hatcheries want to harness maximum growth potential of the juveniles, significantly reduce the culture duration and without compromising physiological competencies to endure post-release hardships in the wild. Feeds beyond conventional carp-based choices which comfortably fulfilled our calculated target nutrient range (even beyond standard requirements for common carp) resulted in best growth and body nutrient reserves. Not only optimum diets, but various pre-conditions around feeding are also important to consider for optimization of captive feeding of these conservation-priority and data-poor fish species. Barbels could be a nutrition model for understanding nutrition and metabolism of other lesser-known native rheophilic cyprinids of the region.

## Figures and Tables

**Figure 1 biology-10-01245-f001:**
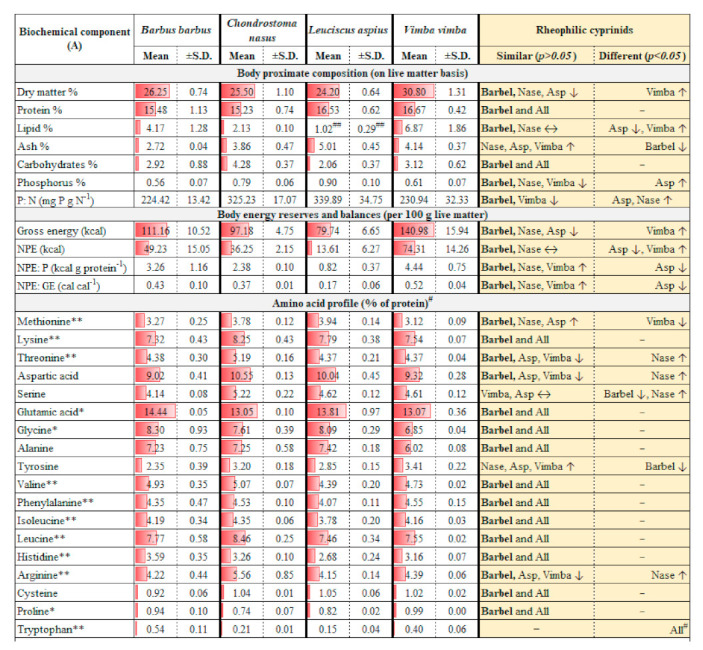
**(A,B): Body composition of four common native rheophilic cyprinids (barbel *Barbus barbus*, nase *Chondrostoma nasus*, asp *Leuciscus aspius* and vimba bream *Vimba vimba*) of Central European drainage systems.** All values are on live matter basis. Abbreviations: P, phosphorus; N, nitrogen; NPE, non-protein energy; NPE: P, non-protein energy to protein ratio; NPE:GE, non-protein energy to gross energy ratio; SFA, saturated fatty acids; MUFA, monounsaturated fatty acids; PUFA, polyunsaturated fatty acids; n-3 FA, omega-3 fatty acids; n-6 FA, omega-6 fatty acids. Red bars show relative richness. Notes for (**A**): * Non-essential but functional amino acids. ** Essential (indispensable) amino acids. # Tryptophan values were non-accredited (≈low confidence). All amino acids results except tryptophan are claimed to be accredited. ## Overall body lipid content quite low across samples (range 0.6–1.4%). Mean and SD re-calculated by ignoring lower quartiles (below 0.7%). Notes for (**B**): ## Vimba bream had insufficient sample matrices left.

**Figure 2 biology-10-01245-f002:**
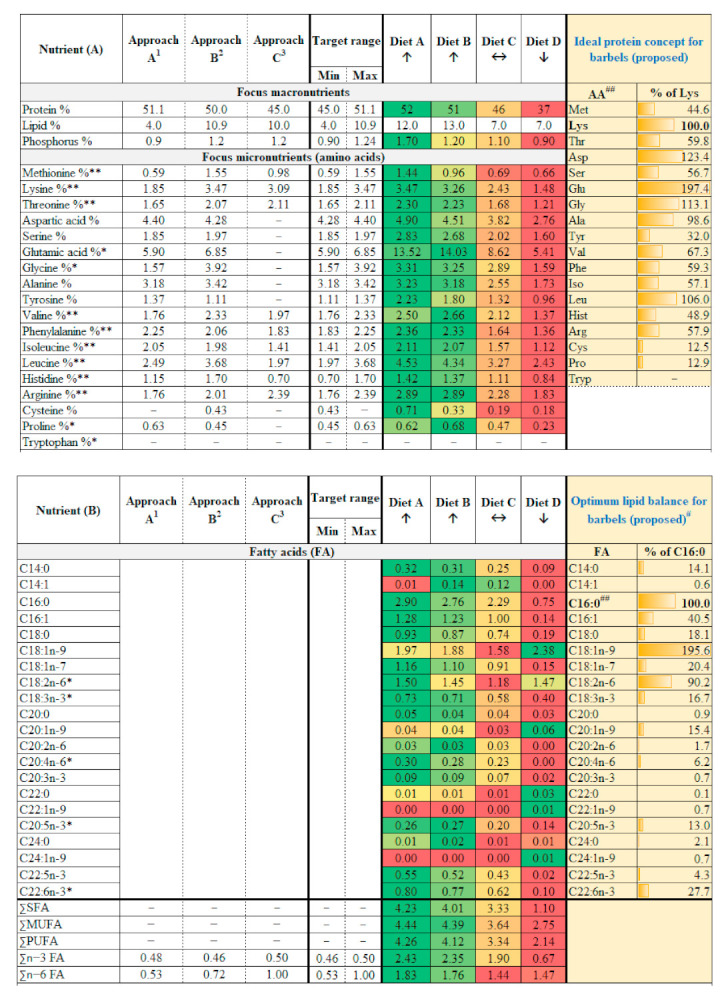
**(A,B): Target nutrient range and experimental diets considered for raising juvenile (0+, 1+) barbel.** Crude nutrient recommendations for commercial diet selection (green data cells only) may be followed in the context of riverine stocking (≈extraordinary size-at-age within shortest possible time in captivity) of barbels and other biochemically related rheophilic cyprinids native to the region. For additional information about the selected diets, refer to Appendix A. Colour scales indicate low (red) to high (green) graded levels of nutrients in our selected experimental diets. Notes for (**A**): ^1^ Chironomids (100% DM basis): protein 56.8%, lipid 4.4%, phosphorus 0.99%, Met+Cys 0.66%, Lys 1.65%, Thr 1.46%, Asp 4.89%, Ser 2.05%, Glu 6.55%, Gly 1.74%, Ala 3.53%, Tyr 1.52%, Val 1.96%, Phe 2.5%, Ileu 2.28%, Leu 2.77%, His 1.28%, Arg 1.95%, Pro 0.7% (Roy et al. 2021). Values multiplied by 0.9 (see, formula in methods). ^2^ Calculated from barbel body composition (Figure 1) and multiplied by cyprinid retention scheme (see, formula in methods). ^3^ Adopted from nutritional specifications for phylogenetic relative, common carp. Values provided are for juvenile carps <20 g body weight. Notes for (**B**): * Essential fatty acids. ^#^ Optimum fatty acids balance proposed in the sense of ‘ideal protein concept’. derived from barbel body lipid composition. ^##^ Palmitic acid (C16:0) was selected as central FA (see discussions). ^1^ Chironomids (100% DM basis): ∑n−3 FA 0.53%, ∑n−6 FA 0.59%. Values multiplied by 0.9 (see, formula in methods). ^2^ Calculated from barbel body composition (Figure 1) and values used as it is (see methods for clarification). ^3^ Adopted from nutritional specifications for phylogenetic relative, common carp. Values provided are for juvenile carps <20 g body weight.

**Figure 3 biology-10-01245-f003:**
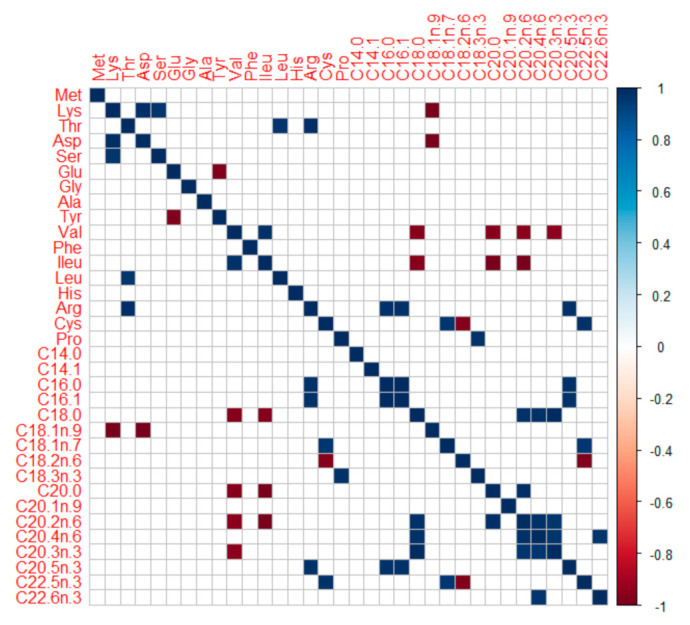
**Pearson’s 2-tailed correlogram (at *p* < 0.05) on associations between mean body amino and fatty acids of some native, large-bodied rheophilic cyprinids of Central European drainage systems (barbel, nase, asp, and vimba bream).** Note the significant associations of long-chain FAs with some consistent AAs (valine, phenylalanine, arginine, and cysteine). Fatty acids notations (dash) are auto-formatted while generating correlogram in RStudio, the standard notations of the same FAs can be found in Figure 1.

**Figure 4 biology-10-01245-f004:**
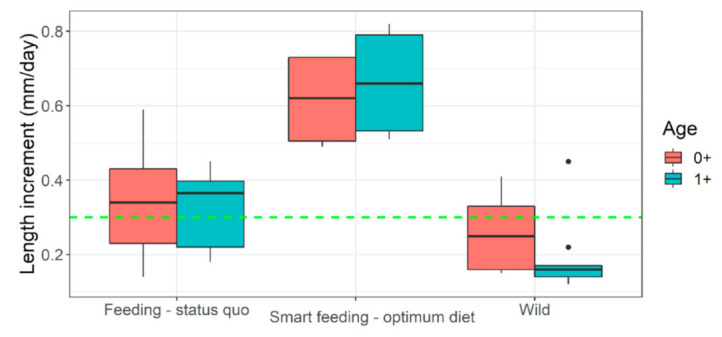
**Length increment in captive (feeding regimen wise) versus wild *Barbus barbus*.** Green horizontal dashed line indicates ‘reasonably good size increment’ (i.e., above median). Black dots indicate some outliers in European river populations. Smart feeding-optimum diet = diets A, B. Feeding-status quo = diets C, D + some previous growth trials (sources in Appendix A). Wild = European rivers metadata (sources in Appendix A).

**Figure 5 biology-10-01245-f005:**
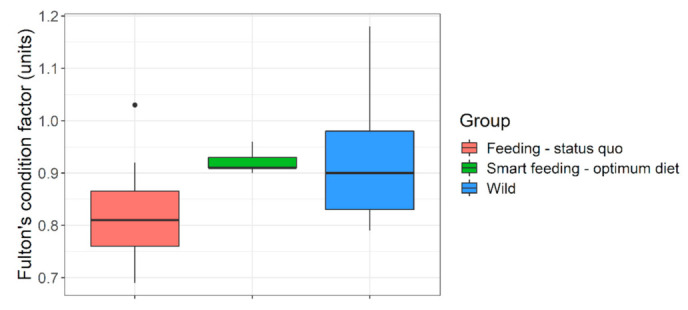
**Body fitness (Fulton’s condition factor) of wild versus captive (feeding regime wise) *Barbus barbus*.** Smart feeding-optimum diet = diets A and B. Feeding-status quo = diets C, D + some previous growth trials (sources in Appendix A). European rivers = wild data (sources in Appendix A).

**Figure 6 biology-10-01245-f006:**
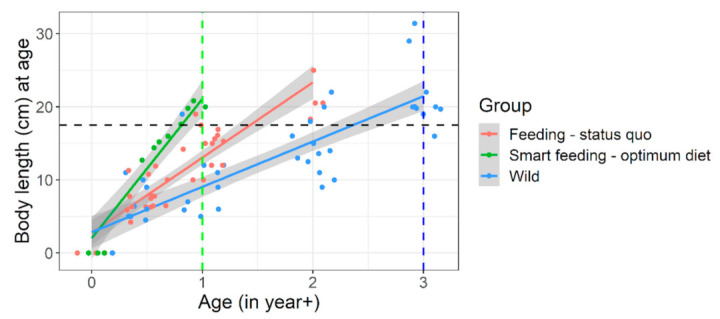
**Growth trajectory or size-at-age of captive (under different feeding regimen) versus wild *Barbus barbus*.** Black horizontal line indicates achievable body length (~20 cm) mostly in wild 3+ barbels which was achieved in early 1+ juveniles (green vertical line) under smart feeding of optimum diet. Smart feeding-optimum diet = diets A, B. Feeding-status quo = diets C, D + previous captive trials (sources in Appendix A). Wild = European rivers metadata (sources in Appendix A). Detailed growth trajectory observed within our experimental framework is provided in Appendix A.

**Figure 7 biology-10-01245-f007:**
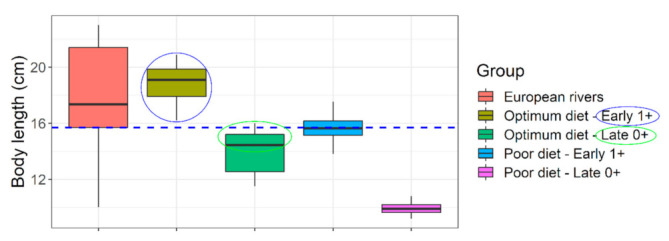
**Size suitability of captive raised *Barbus barbus* under different feeding regimen for river stocking purposes.** Blue horizontal line indicates a representative body length of juvenile barbels in European rivers (=15.7 cm). Colored circles indicate convincing (blue) or nearly convincing (green) stocking size suitability, at a very young age (8–13 months old). European rivers data: (sources in Appendix A).

**Figure 8 biology-10-01245-f008:**
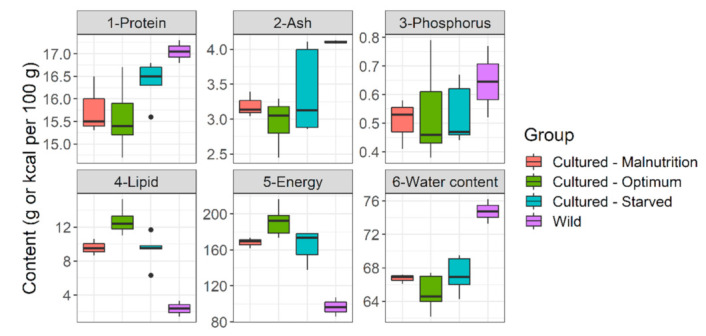
**Whole body reserves (live weight basis) of *Barbus barbus* under different feeding regimens and in the rivers.** Cultured-Malnutrition = barbels fed sub-standard diet D. Cultured-Optimum = barbels fed optimum diets A and B. Cultured-Starved = barbels starved for 60 h previously fed an average diet (diet C). Wild = wild barbels (Czech rivers).

**Figure 9 biology-10-01245-f009:**
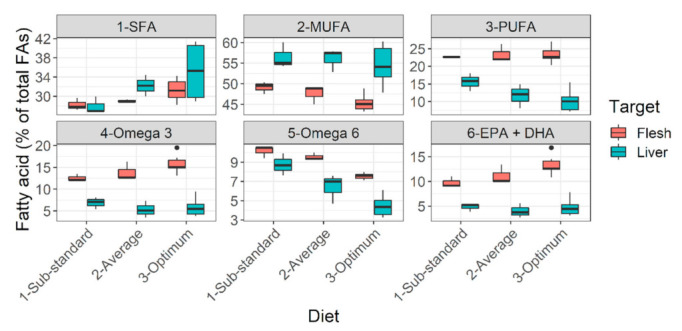
**Lipid deposition pattern of *Barbus barbus* in liver and flesh under different diets.** Optimum diets = diets A, B. Average diet = diet C. Sub-standard diet = diet D. Abbreviations: FAs, fatty acids; SFA, saturated fatty acids; MUFA, monounsaturated fatty acids; PUFA, polyunsaturated fatty acids; EPA + DHA, sum of eicosapentaenoic acid and docosahexaenoic acid. Notice the decreasing trend of essential lipid reserves (PUFA *p* > 0.05, Omega-3 *p* < 0.05, EPA + DHA *p* < 0.05) from optimum to sub-standard diet. May be collated with the color pattern in Appendix A.

**Figure 10 biology-10-01245-f010:**
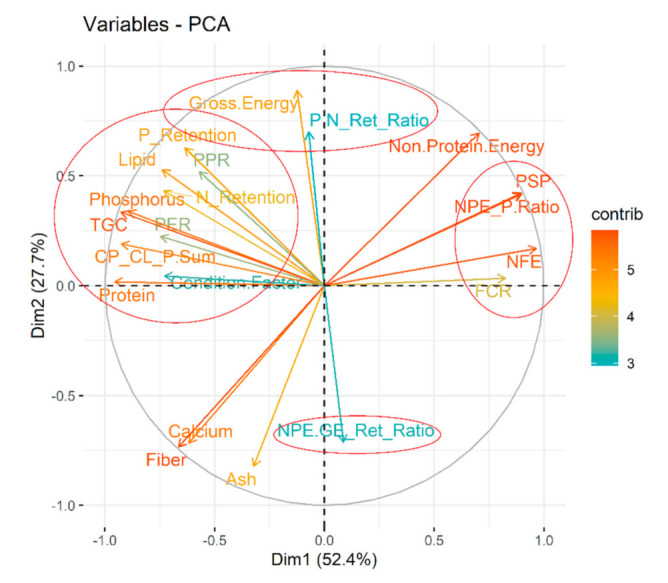
**Nutritional intricacies behind growth of 0+ and 1+ age *Barbus barbus*.** Dimension-1 of the principal component analyses (PCA) explain majority (>50%) of the variability and selected for interpretation. Four related clusters were identified (encircled) relative to growth or thermal growth coefficient (TGC).

**Figure 11 biology-10-01245-f011:**
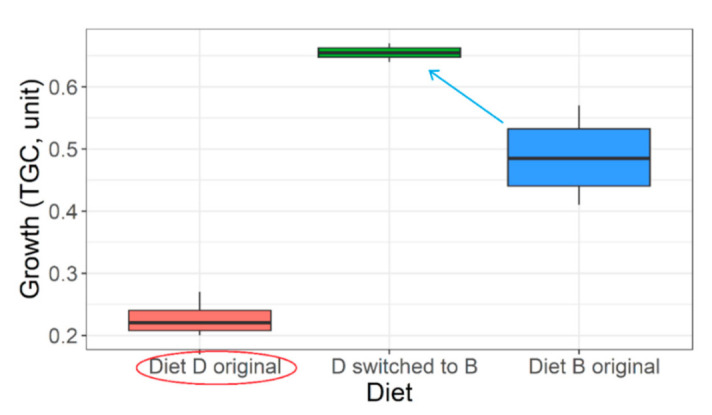
Compensatory growth demonstrated by poorly growing 0+ *Barbus barbus* with sub-standard diet history (red circle) suddenly switched to an optimum diet (fish on diet D switched to diet B). The blue arrow indicates the ‘slingshot effect’ or net compensatory growth (+34.7%) relative to the growth originally demonstrated by diet B (i.e., mid-0+ barbels fed diet B for 100 days; Table 1).

**Table 1 biology-10-01245-t001:** **Growth performance of *Barbus barbus* (initial length 7.7 ± 0.5 cm; initial weight 3.9 ± 0.8 g; 5 months old) during a 100-day growth trial (21.7 °C) under feeding at ~6% of body weight.** Values are expressed in mean ± standard deviation sometimes with interquartile range and coefficient of variation in parentheses. TL, total length (cm); BW, body weight (g); TGC, thermal growth coefficient (units); LI, length increment (mm day^−1^); PER, protein efficiency ratio (g weight-gained per g protein-fed); P:N retention ratio, mg P-stored per g N-retained. NPE: GE retention ratio, cal non-protein energy reserved per cal gross energy retained.

Parameters	Diet A	Diet B	Diet C	Diet D
Morphometrics				
Age (years) *	Late 0+ (8+ months)	Late 0+ (8+ months)	Late 0+ (8+ months)	Late 0+ (8+ months)
Final TL (cm)	15.2 ± 1.1 (14.5–16; 7.1%) ^a^	12.7 ± 1.9 (11.5–14.4; 14.8%) ^b^	11.3 ± 1.1 (10.7–11.9; 9.7%) ^c^	10.0 ± 1.1 (9.2–10.8; 10.8%) ^d^
Final BW (g)	34.4 ± 8.1 (29.0–41.7; 23.5%) ^a^	20.0 ± 9.7 (13.7–23.8; 48.3%) ^b^	13.3 ± 4.1 (11.0–14.9; 30.7%) ^c^	8.9 ± 3.0 (7.2–11.2; 33.5%) ^d^
CF (units)	0.96 ± 0.05 ^a^	0.92 ± 0.10 ^b^	0.91 ± 0.06 ^b^	0.86 ± 0.07 ^c^
**Growth indicators**				
TGC (unit)	0.77 ± 0.04 ^a^	0.49 ± 0.06 ^b^	0.36 ± 0.02 ^c^	0.23 ± 0.03 ^d^
LI (mm day^−1^)	0.74 ± 0.03 ^a^	0.51 ± 0.08 ^b^	0.35 ± 0.02 ^c^	0.23 ± 0.04 ^d^
Yield (kg m^−3^ day^−1^)	0.23 ± 0.01 ^a^	0.15 ± 0.03 ^b^	0.08 ± 0.016 ^c^	0.04 ± 0.007 ^d^
**Feed utilization**				
Protein retention (%)	39.2 ± 0.6 ^a^	26 ± 1.6 ^a,b^	22.6 ± 0.7 ^b^	23.3 ± 1.9 ^b^
Lipid retention (%)	117.2 ± 3 ^a,b,#^	95.5 ± 5.7 ^b,#^	169.4 ± 2.7 ^a,#^	77.4 ± 8.2 ^b^
Phosphorus retention (%)	53.3 ± 2.3 ^a^	35.5 ± 3.8 ^b^	22.3 ± 1.7 ^c^	32.2 ± 3 ^b^
**Physiological performance markers**				
PER	0.80 ± 0.01 ^a^	0.65 ± 0.04 ^b^	0.59 ± 0.01 ^b,c^	0.52 ± 0.04 ^c^
P: N retention ratio	274.3 ± 8.0 ^a^	200.5 ± 9.4 ^b^	149.3 ± 15.4 ^c^	217.2 ± 18.8 ^b^
NPE: GE retention ratio	0.63 ± 0.01 ^a^	0.71 ± 0.01 ^b,c^	0.74 ± 0.01 ^b^	0.65 ± 0.01 ^a,c^
**Feed economics**				
Feed cost per kg yield	9.6€	8.4€	5.36€	6.7€

^a, b, c, d^ Superscripts denote statistically different (*p*  <  0.05) groups. * Age since onset of exogenous feeding. ^#^ Occurrence of de-novo lipogenesis.

**Table 2 biology-10-01245-t002:** **Growth performance of *Barbus barbus* (initial length 14.5 ± 2.2 cm; initial weight 29.1 ± 13.8 g; 11 months old) during a 64-day ‘validation’ trial (22.8 °C) under feeding at ~4% body weight).** Approaches used (diet calculation, selection; Figure 2) and trends observed among diets (despite age difference) successfully validated as they matched with patterns in Table 1.

Parameters	Diet A	Diet B	Diet C	Diet D
Morphometrics				
Age (years) *	Early 1+ (13+ month)	Early 1+ (13+ month)	Early 1+ (13+ month)	Early 1+ (13+ month)
Final TL (cm)	19.8 ± 2.1 (18.7–20.8; 10.5%) ^a^	17.6 ± 2.5 (16.2–20; 14.5%) ^b^	16.9 ± 2.9 (16.8–19; 17.3%) ^b,c^	15.6 ± 2.6 (13.8–17.5; 16.7%) ^c^
Final BW (g)	73 ± 23.1 (57.8–81.3; 31.7%) ^a^	52.6 ± 20.7 (39.6–68.1; 39.4%) ^b^	46 ± 24.3 (27.3–57.7; 52.8%) ^b^	33.7 ± 17.9 (32.2–45.7; 53.1%) ^c^
CF (units)	0.90 ± 0.06 ^a^	0.91 ± 0.08 ^a^	0.87 ± 0.06 ^b^	0.81 ± 0.06 ^c^
**Growth indicators**				
TGC (unit)	0.74 ± 0.04 ^a^	0.49 ± 0.05 ^b^	0.33 ± 0.03 ^c^	0.14 ± 0.04 ^d^
LI (mm day^−1^)	0.81 ± 0.02 ^a^	0.55 ± 0.03 ^b^	0.39 ± 0.02 ^c^	0.21 ± 0.03 ^d^
Yield (kg m^−3^ day^−1^)	0.16 ± 0.01 ^a^	0.10 ± 0.02 ^b^	0.06 ± 0.013 ^c^	0.02 ± 0.008 ^d^
**Feed utilization**				
Protein retention (%)	36.8 ± 3.8 ^a^	18.9 ± 4.8 ^a,b^	11.9 ± 3.4 ^b,c^	7.9 ± 1.4 ^c^
Lipid retention (%)	121.5 ± 8 ^a,#^	94.9 ± 12.2 ^a,#^	169.2 ± 16.6 ^b,#^	36 ± 7.8 ^c^
Phosphorus retention (%)	56.8 ± 4.3 ^a^	25.8 ± 7 ^a,b^	10.4 ± 5.6 ^b^	15.8 ± 6.3 ^b^
**Physiological performance markers**				
PER	0.76 ± 0.06 ^a^	0.57 ± 0.09 ^b^	0.44 ± 0.06 ^b^	0.25 ± 0.05 ^c^
P: N retention ratio	313.4 ± 10.0 ^a^	200.1 ± 8.1 ^a^	148.9 ± 23.2 ^b^	295.8 ± 137.9 ^a^
NPE: GE retention ratio	0.65 ± 0.01 ^a^	0.78 ± 0.03 ^b^	0.84 ± 0.03 ^c^	0.74 ± 0.02 ^b^
**Feed economics**				
Feed cost per kg yield	10€	9.8€	7.2€	14.8€

^a, b, c, d^ Superscripts denote statistically different (*p*  <  0.05) groups. * Age since onset of exogenous feeding. ^#^ Occurrence of de-novo lipogenesis.

## Data Availability

All data are provided in the article. Data will be made available on request.

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
