# Peer review of "Understanding Nutrition and Metabolism of Threatened, Data-Poor Rheophilic Fishes in Context of Riverine Stocking Success- Barbel as a Model for Major European Drainages?"

_biology, 2021, doi:10.3390/biology10121245_

Round 1

Reviewer 1 Report

Thanks to the authors for their answers and corrections.

Reviewer 2 Report

The authors provided a full revision of the manuscript, increasing its quality.

This manuscript is a resubmission of an earlier submission. The following is a list of the peer review reports and author responses from that submission.

Round 1

Reviewer 1 Report

The paper approaches a very interesting topic about the role of feeding strategies during rearing rheophilic fish prior to be released into the natural environment. Not only the opportunity, but also the quality of the content, make it worth being published. However, some questions need to be solve.

To determine the parameters of use of feed utilization, it is necessary to know the feed intake. What was the total feed intake per fish during the experimental period in each trial? The authors estimate the feed intake according to the daily feeding, providing the feed ration manually to apparent satiety (in the second trial, because in the first trial it is ad libitum; this is an error since ad libitum is not what same as apparent satiety). The authors mention that fish were under visual supervision, ensuring complete utilization of administered feed. I can accept this without recovering the uneaten food. But it is impossible for fish fed apparent satiety to eat the same daily ration in all tanks when the diets were made with different ingredients and/or different inclusion levels.

Tables 3 and 4 show de novo lipogenesis in fish fed diets A and C. In the discussion, the authors describe the role of some amino acids in fatty acid metabolism. Table 2A shows the amino acid content in the tested diets. How could it be explained that the fish fed diet B have not manifested de novo lipogenesis when their amino acid profile is, for all the amino acids described, in-between between diet A and C?

In line 431, the authors state that protein retention was higher with optimal diets. But neither Table 3 nor Table 4 show superscripts for this or other food utilization parameters to denote statistically significant differences. Authors should describe the statistical analysis in a section within the Material and Methods. Thus, with the data of the three tanks per diet, they can make a simple means comparison and obviously detect significant differences.

Reviewer 2 Report

Major comments:

  • First, the manuscript needs to be massively edited by a native English speaker to improve the language of the MS and fix numerous errors. I mentioned some of them, but still, more works needed to be done. Many parts of this MS are unclear and is worsened by some unnecessary complicated uncommon phrases. I suggest writing simple, short, and clear. Understanding many parts of the MS because of these issues is hard.
  • The supplementary file is confusing and has a discussion and conclusion section and some repetitive parts. Please make sure you provide only the necessary information.
  • Presenting the results was also so confusing as they provided too much information.
  • They provided 119 references which is too much. Most parts of the MS should be summarized.

However, I have touched on some more points that can contribute to the improvement of this MS.

Minor comments

Abstract

  • I suggest writing a Simple Summary and abstract more straightforward. Make sure any phrase you use, such as conventional nutrition, imperilled rivers, cohorts, metabolically better insured for post-stocking hardships, etc., has been used correctly. Please revise these two sections.
  • Line 10-12, please revise it. It has already been worked, and no need to re-think.
  • Line 14, confusing; please revise it.
  • Line 15, this sentence needs a reference. Please delete it.
  • Line 16-17, please revise it.
  • Line 18, please revise it.
  • Line 18-21, is not novel and also revise this sentence. Try to write simple and avoid complicating the meaning by using some unmatched words.
  • Line 20-21, confusing again.
  • Line 25-29, please revise it and try to write it simpler.
  • Line 30-35, please revise it.
  • Line 35, “nutrition decisions” is not an English phrase; please revise it.
  • Line 35, are discussed or was discussed
  •  
  • Introduction:
  • Line 42-45, please revise it, “Rheophilic specialists”?
  • Line 57-58, poorly connected to the last sentence; please revise it.
  • Line 59, Restocking or stocking?
  • Line 60-66, please revise this part.
  • Line 70-72, please revise is, do you mean “captive feeding”? I think in many parts of this MS, you wrote nutrition incorrectly.
  • Line 75-78, please revise it.
  • Line 87, “are not well understood” and not standardized.
  • Line 87-89, is not clear; please revise it. Why is strong background important?
  • Line 92-93, please revise it; I think you mean in terms of protein contents.
  • Line 95, you mean nutritional physiology. Nutritional demands?
  • Line 101-104, please revise it.
  • Material and methods
  • Well-organized section. Clear fellow and all required details were provided.
  • Please mention how many percentages of water were exchanged each day if you have monitored.
  • Line 122, please revise it.
  • Line 131, in the house!!!, please revise it. Also, you need briefly explain the amino acids and fatty acid analysis process.
  • Line 132-139, confusing part, is not clear why did you use species-wise. Amino acids and fatty acids data are usually normal. Please clearly state which data was not normal, as can be because of a mistake in calculation or analyzing data.
  • Line 141-144, is not clear; please revise it.
  • Line 150 and elsewhere, please make sure you used nutrients and nutrients correctly.
  • Line 161-162, confusing, is not clear.
  • Line 164-170, please add the protein, lipid, ash and energy levels of these diets.
  • Line 185, revise old food.
  • Line 201-205, is not clear why diet D and C were worst-performing diet group. Adding proximate composition will make it clear.
  • Line 214, confusing; please revise it.
  • Line 217-219, is not clear what you did here; if there is apparent satiation, this 4% does not make sense—no need to calculate apparent satiation.
  • Line 219-227, is not clear how did you calculate FCR. Was it apparent satiation? 3-4%, 6%?.Please deeply revise this section.
  • Line 233-236, please revise this part.
  • Line 246, 254, please make sure you used the correct phrases. Please try to write it simpler.

Results

  • Well-written section, all necessary things have been covered.
  • Line 258-264, is not clear what do you mean by was comparable? Comparable with what?
  • Line 269, do you have any reference that CV>12% can be considered high?. From my idea, even until 30% variation in this data is normal in fish and sentence 269, can be doubtful.
  • Line 271, please write clear. Please name the four species and delete “Detailed.”
  • Table 1, lipid content of 0.6% seems wrong. I strongly suggest double-checking all data here.
  • Table 1, please explain what the red and yellow colours is.
  • In table 1, you can delete the fatty acids that do not have data for all four species.
  • Line 282 and elsewhere, please write is clear, and avoid this style of writing. Please clearly state which unsaturated fatty acids.
  • Line 282-283, is not clear.
  • Line 289, please revise it.
  • Line 290, is not clear what do you mean by “Since barbel was a representative species in terms of body composition”.
  • Table 2, Ideal protein concept for barbels & related rheophilic cyprinids (crude)# is not clear; please explain there.
  • Please notice that amino acids and fatty acids are daily variations and also different in seasons. When you discuss your results, please consider this issue.
  • Line 316-320, please revise it. Please make sure you used fish and fishes correctly through the MS.
  • Table 3 and other Tables and Figures, please make sure you provided all information in footnotes. Each Table should be clear without required checking from other parts of the MS. For example, is not clear what do you mean by optimum diet, average and sub-standard diet.
  • Line 357-366, please revise this part.
  • I could not understand how you designed the poor diet. If you knew is a poor diet based on which logic you fed fish with this diet? Please clearly explain in M&M.
  • Line 394-395, please revise this part.
  • Please mention the optimum nutrients for this fish species in the abstract. For example, lipid and protein, EPA, DHA, n-3 LC-PUFA and amino acids.
  • Line 399-410, is it for the result section or discussion?
  • Line 411-455, please summarize this part, too much information was provided. Please only mention important results.
  • Line 414, “Quite obviously” is not formal and scientific
  • Line 414, too much general, how responded?
  • Line 421, what was found between x and y??
  • Line 411-455, is it for result or discussion? Please make sure you did not explain too much in the result section.
  • Line 454, please revise it and make sure you use the right verbs and scientific terms.

Discussion

  • I suggest you strongly write the Result and Discussion with each other. Your style of writing is suited to this style.
  • Line 482, how much was for smart feeding? Throughout the MS, please make sure you explained clearly and suggested a clear and straightforward dosage of nutrients
  • Line 488, I do not understand how a salmonid feed can be status quo feeding for cyprinids?. Please explain well.
  • Line 499, is not clear; please explain.
  • Line 517-523, please revise it.
  • Line 546-554, please revise this part, poor English and unclear.
  • Line 551-554, why? Is not clear.
  • Line 561, please make sure you defined the abbreviations for the first time in the MS.
  • All parts of the discussion are required a deep revision. Please try to summarize the information and make a clear route to the conclusion.
  • Line 587, protein synthesis??
  • Line 622-629, please revise it; unclear.
  • Line 639-689, please revise it and make sure you explain simply and clearly. Please avoid using complicated phrases and sentences.
  • Line 691-692, is not clear.
  • Please mention the nutrients requirements of this fish species based on your discovery.
  • Please summarize the MS and references and provide a maximum of 90 references.
  • The supplementary file is confusing has it has a discussion and conclusion section and some repetitive parts. Please make sure you provide only the necessary information.

Kind regards

Reviewer 3 Report

The authors presented a complete and exhaustive evaluation of nutrition importance to rheophilic fish. This work provides a good discussion about available data and provides bases for further research in the area.